# Silymarin Reduced Insulin Resistance in Non-Diabetic Women with Obesity

**DOI:** 10.3390/ijms25042050

**Published:** 2024-02-08

**Authors:** Karla MacDonald-Ramos, Adriana Monroy, Mariana Bobadilla-Bravo, Marco Cerbón

**Affiliations:** 1Programa de Doctorado en Ciencias Biomédicas, Universidad Nacional Autónoma de México, Ciudad de México 04510, Mexico; 2Facultad de Química, Universidad Nacional Autónoma de México, Ciudad de México 04510, Mexico; ladiredemariana@comunidad.unam.mx; 3Servicio de Oncología, Hospital General de México Dr. Eduardo Liceaga, Ciudad de México 06720, Mexico; adriana.monroy@salud.gob.mx

**Keywords:** insulin resistance, silymarin, obesity, type 2 diabetes, prediabetes

## Abstract

Silymarin has ameliorated obesity, type 2 diabetes (T2DM), and insulin resistance (IR) in combination with standard therapy, diet, or exercise in recent studies. Obesity and IR are the main risk factors for developing T2DM and other metabolic disorders. Today, there is a need for new strategies to target IR in patients with these metabolic diseases. In the present longitudinal study, a group of non-diabetic insulin-resistant women with type 1 and type 2 obesity were given silymarin for 12 weeks, with no change in habitual diet and physical activity. We used the Homeostatic Model Assessment for Insulin Resistance Index (HOMA-IR) to determine IR at baseline and after silymarin treatment (t = 12 weeks). We obtained five timepoint oral glucose tolerance tests, and other biochemical and clinical parameters were analyzed before and after treatment. Treatment with silymarin alone significantly reduced mean fasting plasma glucose (FPG) and HOMA-IR levels at 12 weeks compared to baseline values (*p* < 0.05). Mean fasting plasma insulin (FPI), total cholesterol, high-density lipoprotein cholesterol (HDL-C), low-density lipoprotein cholesterol (LDL-C), triglycerides (Tg), indirect bilirubin, and C-reactive protein (CRP) levels decreased compared to baseline values, although changes were non-significant. The overall results suggest that silymarin may offer a therapeutic alternative to improve IR in non-diabetic individuals with obesity. Further clinical trials are needed in this type of patient to strengthen the results of this study.

## 1. Introduction

Each year, around 2.8 million individuals worldwide die from causes associated with overweight or obesity [1,2,3]. Obesity is the excess or abnormal accumulation of adipose tissue (AT) in the body, which is harmful to health and linked to the development of metabolic disorders, among others [4,5,6,7]. Obesity is also a systemic and multifactorial disease and is considered an inflammatory, chronic, and low-grade systemic process, mainly due to the infiltration of macrophages that permeate AT, modulate inflammation, and are the principal source of proinflammatory cytokines. As proinflammatory cytokines interfere with insulin signaling pathways, the inflammatory state generated in obesity is a central factor in developing insulin resistance (IR) [5,6,7,8,9,10]. Specific medications for glycemic control can optimize insulin response and reduce insulin demand. These include metformin, glucagon-like peptide inhibitors, sodium-glucose cotransporters-2, dipeptidyl peptidase-4 cotransporters, and thiazolidinediones. Regardless of drug treatment, when IR is chronic, it can progress to the development of metabolic syndrome, non-alcoholic fatty liver disease (NAFLD), and type 2 diabetes (T2DM) [11,12,13,14].

Other strategies employed in treating metabolic diseases include natural extracts, such as silymarin [15,16,17], combined with standard treatment for metabolic disease in several studies [18,19,20,21,22,23,24,25,26,27]. Silymarin contains six flavonolignans, primarily silybin A and B, obtained through extraction [18,19,20,28]. The beneficial effects of silymarin on human health, including its antioxidant, anti-inflammatory, antiviral, cytoprotective, and anticarcinogenic effects, have been studied [21,22,23,25,26,29,30]. Notably, silymarin has ameliorated liver disease, as shown in in vitro and animal studies [31,32,33,34,35,36,37,38,39,40,41,42,43,44] and in humans [45,46,47,48]. Researchers have developed different pharmacological formulations of silymarin due to its low bioavailability in its natural state; new biotechnologies, including nanotechnologies, have solved this problem [28,49,50,51,52].

Interestingly, recent studies have used silymarin to help treat individuals with metabolic disease and IR. Many authors have reported the effects of silymarin administration on IR in several diabetes, obesity, and NAFLD animal study models and in vitro study models. In addition to an overall reduction in IR, silymarin treatment also reduced weight and epididymal fat mass, improved endothelial dysfunction, decreased oxidative stress indicators, such as nuclear factor kappa-light-chain-enhancer of activated B cell (NF-kB expression and tumor necrosis factor-alpha (TNF-α) levels, and restored, or prevented the inhibition of, the insulin receptor substrate 1(IRS-1)/phosphatidylinositol-3 kinase (P13K)/protein kinase B (Akt) pathway, and others [42,43,53,54,55,56,57,58,59,60,61]. In clinical studies, silymarin combined with standard treatment and other natural compounds, such as berberine and vitamins, improved IR, lipid metabolism, or inflammatory markers in T2DM, alone or with comorbidities, such as hepatic cirrhosis, chronic liver diseases, obesity, or dyslipidemia [44,62,63,64,65,66,67,68,69,70,71,72,73,74,75,76,77,78,79,80,81,82,83]. Recently, a detailed meta-analysis of the metabolic effect of the association of silymarin and berberine in clinical trials found that the coadministration of these nutraceuticals improved lipid and glucose profiles and may promote cardiometabolic health. Other recent clinical studies with silymarin and novel nutraceutical supplements improved biomarkers associated with obesity, or lipid metabolism, but not IR [84,85,86].

These studies show that silymarin administration improved various parameters associated with obesity and T2DM. However, it is unknown whether silymarin treatment alone decreases IR in non-diabetic individuals with obesity. This study aimed to establish whether silymarin treatment alone ameliorated IR in obese individuals without T2DM.

## 2. Results

After 12 weeks of silymarin treatment, we observed no significant differences in body composition parameters in the patients in the patients. We show the clinical, anthropometric, and body composition characteristics of the enrolled patients in Table 1.

We observed a reduction in almost all biochemical parameters after treatment was completed compared to baseline values (Table 2). Mean fasting plasma glucose (FPG) and Homeostatic Model Assessment for Insulin Resistance Index (HOMA-IR) levels decreased significantly after silymarin treatment (t = 12 weeks) (Appendix A). Mean fasting plasma insulin (FPI), total cholesterol (TC), high-density lipoprotein cholesterol (HDL-C), low-density lipoprotein cholesterol (LDL-C) and triglycerides (Tg), indirect bilirubin, and C-reactive protein (CRP) levels decreased after 12 weeks of silymarin treatment, although these differences were non-significant.

In the oral glucose tolerance test (OGTT), we observed a significant decrease in mean FPG levels after silymarin treatment (t = 12 weeks) compared to baseline values. Additionally, mean glucose levels at 30, 60, 90, and 120 min, and insulin levels at 0, 30, 60, and 120 min, decreased after silymarin treatment (t = 12 weeks) compared to baseline values, although the change was non-significant. Interestingly, mean insulin levels increased at 90 min in the OGTT after silymarin treatment (t = 12 weeks) compared to mean baseline insulin values (Table 3). We show the differences in mean glucose and insulin levels in patients during the OGTT, at baseline, and after silymarin treatment (t = 12 weeks) in Table 3.

The glucose response curves at baseline and after silymarin treatment (t = 12 weeks) were monophasic and followed a similar pattern [87,88], displaying maximum mean glucose values at 60 min. Mean glucose baseline values two hours post-glucose load during the OGTT indicated prediabetes. However, mean glucose values after silymarin treatment (t = 12 weeks) two hours post-glucose load during the OGTT indicated normoglycemia (Appendix A) [89,90,91,92]. By contrast, the insulin clearance curves at baseline and after silymarin treatment (t = 12 weeks) did not follow a similar pattern. The baseline insulin clearance curve was biphasic [87,88]; mean insulin levels increased until 60 min, then decreased until 90 min, and increased again with maximum values at 120 min. After 12 weeks of silymarin treatment, the insulin clearance curve was monophasic [87,88]; mean insulin values gradually increased until 90 min, reached maximum values, and decreased slightly until 120 min (Appendix A). We show the glucose response and insulin clearance curves during the OGTT, at baseline, and after silymarin treatment (t = 12 weeks) in Appendix A.

## 3. Discussion

One of the aims of this study was to analyze the effect of silymarin treatment alone on IR in obese individuals without T2DM. The silymarin formulation used in this study improved IR significantly in the group of non-diabetic women with obesity after twelve weeks of treatment. Primarily, the results showed that silymarin administration reduced mean glucose and HOMA-IR levels significantly, as in other clinical trials that analyzed the effect of silymarin treatment in insulin-resistant individuals [62,63,65,66,67,68,69,70,71,72,75,76,80,81,82,93]. A significant reduction in mean glucose levels in our study suggests that silymarin treatment increased insulin sensitivity in target tissues, thereby promoting the expression of the glucose receptor and the entry of glucose into cells. In addition, we observed this reduction employing a lower dose of silymarin and with a shorter duration of treatment than in other studies that reported a decrease in glucose levels [48,62,65,70,71,72,79,93] or in others that reported a decreased IR [64,65,66,67,68,69,73,74,76,80,81,82,83,93].

The results of our study are in line with the results of other studies that also reported a decrease in IR in diabetic individuals with silymarin administration as an adjunct to standard insulin therapy for T2DM [65,67,69] or in diabetic individuals with cirrhosis [62,63]. Other studies that reported a decrease in IR employed silymarin combined with *B. aristata* to treat patients with T2DM simultaneously with insulin/metformin/incretins/sulfonylureas/glitazones [66] or without standard treatment for T2DM but with a hypocaloric diet and physical activity [68]. Other authors that reported a decrease in IR conducted studies in individuals with NAFLD, with or without hepatitis C virus (HCV), but all with silymarin combined with vitamin (Vit) E [70,71,72,74], Vit E and the Mediterranean diet or lifestyle modifications [75,76], or with Vit D [79,93]. Our study’s population most closely resembled those in three studies conducted in groups of dyslipidemic overweight individuals, where two of the three studies reported a decrease in IR [81,82,83,94]. However, treatment with silymarin in these studies was in combination with *B. aristata*, in addition to the prescription of physical activity, a hypocaloric diet, or a reduced dose of statins. In our study, we did not use silymarin combined with other nutraceuticals or prescribe a change in diet or physical activity to analyze the effects of silymarin alone on IR, thus avoiding the presence of confounding factors in the results of our study. To the best of our knowledge, the present study is the first to use and report that treatment with silymarin alone significantly reduced IR in a group of IR non-diabetic obese women without dietary therapy or physical activity after only 12 weeks of treatment.

Although mean baseline FPG levels in our population decreased significantly after silymarin treatment (t = 12 weeks) (96.7 ± 4.2 mg/dL vs. 91.2 ± 6.8 mg/dL, *p* = 0.041), the values were within a normoglycemic range [89,90,91,92]. However, mean baseline glucose levels at two hours post-glucose load during the OGTT (147.0 ± 21.5 mg/dL) indicated that our study population was prediabetic [89,90,91,92]. After 12 weeks of silymarin treatment, mean glucose levels at two hours post-glucose load during the OGTT (138.6 ± 26.3 mg/dL) were within normoglycemic values. This change suggests that silymarin treatment may have decreased the risk or delayed development of T2DM in our study population by modifying an initial state of prediabetes to normoglycemia [89,90,91,92]. Additionally, the shape of the glucose response curve during the OGTT at baseline and after silymarin treatment (t = 12 weeks) in the results was monophasic. Several studies have reported a monophasic glucose response curve in individuals who have lower insulin sensitivity, decreased β-pancreatic cell function, and may be at an increased risk of developing impaired FPG or T2DM [87,88,95,96,97,98]. This evidence suggests that our study population may share these characteristics since HOMA-IR, used to measure IR in our study, is a simple surrogate index for insulin sensitivity [99,100]. Alternatively, in the results, the shape of the insulin clearance curve during the OGTT at baseline was biphasic, and after silymarin treatment (t = 12 weeks) was monophasic. Several studies have reported both types of insulin clearance curves in individuals with decreased insulin sensitivity [87,88,95,96,97,98]. This evidence further supports that our study population had reduced sensitivity to insulin at baseline.

A decrease in mean FPI levels after silymarin treatment (t = 12 weeks) compared to baseline values was observed, as in other studies [62,63,66,67], although the change was non-significant. A reduction in FPI levels suggests that the demand for insulin production decreased and tissue sensitivity to insulin increased [11]. The absence of a significant decrease in mean FPI levels after silymarin treatment (t = 12 weeks) may be due to the small sample size and duration of silymarin treatment in our study. Additionally, we observed a reduction in mean CT, LDL-C, Tg, BI, and CRP levels, as reported by others, although without significant changes [64,65,66,67,68,69,73,74,76,80,81,82,83,93]. Yet, mean Tg baseline levels in our study population (154.8 ± 47.6 mg/dL) indicated the presence of hypertriglyceridemia [101]. After 12 weeks of silymarin treatment, mean Tg levels (145.0 ± 43.5 mg/dL) no longer displayed hypertriglyceridemia. Therefore, although the decrease was non-significant, hypertriglyceridemia was eliminated in the patients after 12 weeks of treatment with silymarin. Alternatively, we did not observe changes in body composition in our study population after treatment with silymarin. Other studies with silymarin treatment reported a decrease in body mass index (BMI) and body fat [68,70,72,76,77], although the authors prescribed silymarin in combination with physical activity and dietary therapy or with a longer duration of treatment than in the present study. The absence of observable changes in body composition in the present study may be due to the length of treatment and because we did not prescribe any changes in diet or physical activity.

It may be worth mentioning that the few clinical trials that reported a decrease in IR did not do so with BMI values in their population [64,65,73]. We believe it may be helpful to consider the parameter of body and abdominal fat or BMI in clinical trials that analyze the effect of silymarin administration on IR in obese/overweight populations since adipose tissue in overweight or obese states exerts inflammatory effects that alter insulin signaling and induce IR [102,103,104]. Importantly, individuals with IR are at high risk of developing prediabetes and T2DM [11,105]. Yet, clinicians do not routinely measure insulin levels, partly because clinical guidelines do not integrate measures of IR, and other methods for diagnosing prediabetes and T2DM are used [11,89,105]. Similarly, clinicians do not routinely perform the OGTT. In our study, IR and prediabetes were present; the latter was detected at two hours post-glucose load during the OGTT and not through FPG levels, which were in the normal range. Therefore, IR and prediabetes may go undetected by usual methods of diagnosis in a population at risk. We believe it would be advantageous if clinicians employed several tests to detect IR, prediabetes, or T2DM, especially in the at-risk population, since individuals may have undetected hyperglycemia. Lifestyle changes are the general recommendation to prevent or delay the onset of T2DM after IR is detected. It is important to note that although HOMA-IR baseline levels decreased significantly in the present study after 12 weeks of treatment (3.6 ± 1.2 vs. 2.8 ± 0.9, *p* = 0.048), IR was still present in our study population, according to the IR inclusion criteria employed (HOMA-IR ≤ 2.6). A lengthier treatment may have allowed for a further decrease in IR.

One of the main limitations of the present study is the small sample size, which may have prevented significant differences from being observed in other parameters. Another possible limitation is the inclusion of only female patients in our study sample, as some studies have reported sex-dependent differences in response to silymarin treatment [106,107]. Furthermore, we did not stratify the population in our study according to the female reproductive cycle; some individuals may respond differently to silymarin treatment according to hormone levels within the female reproductive cycle and during menopause [106,108,109,110]. Menopause typically occurs between the ages of 47 and 49 in Mexican women, yet the criterion for premature menopause is 40 or less [111]. Therefore, some study participants may have been perimenopausal or transitioning to menopause. In addition, we did not determine HOMA-IR according to the menstrual cycle, and slight variations in IR may have occurred according to the menstrual phase to which premenopausal women were transitioning. Estradiol and progesterone are positively associated with IR [112], and the levels of these hormones vary during the menstrual cycle. A final possible limitation of the present study was the length of treatment. Even though we observed significant changes in IR after silymarin treatment (t = 12 weeks), a lengthier treatment may have allowed for observable differences in other parameters. Indeed, other studies reported a significant improvement in IR after 90 days or three months. However, in these studies, silymarin was given together with standard T2DM treatment [67,69], combined with *B. aristata* in T2DM patients [113], or combined with Vit E in HCV or NAFLD patients [74].

In individuals with an IR diagnosis, lifestyle modifications that include dietary therapy and physical activity are recommended to reduce visceral and body fat and sensitize muscle tissue to insulin [11,114,115]. Currently, there are no medications to treat IR alone, and pharmacological treatment for glycemic control can lead to undesirable side effects. Treatment with silymarin may be an alternative to help reduce IR caused by low-grade inflammation found in obesity [11,20,116]. A decrease in IR may help to prevent or delay the onset of T2DM, which is often reached within ten to fifteen years after IR appears [11]. The results showed that treatment with silymarin significantly decreased IR in a population of non-diabetic women with obesity and IR, which may have contributed to delaying the onset of T2DM. However, more clinical studies in IR-obese individuals with silymarin alone or with other interventions, such as changes in diet and physical activity, are needed to strengthen the results of this study.

## 4. Materials and Methods

### 4.1. Inclusion and Exclusion Criteria for Patient Recruitment

The current pilot study was authorized and performed at the Hospital General de México, Dr. Eduardo Liceaga in Mexico City (registry number D1-15-UME-04-066). The recruitment of patients took place between 2018 and 2020. We recruited six female patients from the Clínica de Atención Integral al Paciente con Obesidad y Diabetes who underwent the OGTT two weeks or less before enrollment. The inclusion criteria were willingness to participate in this study, age 18–45 years, with a BMI of 30–39.9 according to WHO diagnostic criteria, and a HOMA-IR ≥ 2.6 [117,118,119]. Exclusion criteria were patients with T2DM, dyslipidemia requiring pharmacological treatment, alcoholism or drug addiction, chronic infectious diseases, the Human Immunodeficiency Virus, chronic hepatitis, systemic arterial hypertension, auto-immune or chronic inflammatory diseases, who ingested antioxidant multivitamin supplements or nonsteroidal anti-inflammatory drugs 15 days before enrollment, or who were undergoing pharmacological treatment with monoamine oxidase (MAO)-inhibiting antidepressants, and smokers.

### 4.2. Metabolic Intervention

All patients were given silymarin (Neocholal-S^®^, Italmex Pharma, Ciudad de México, Mexico) and asked to take one capsule twice daily for twelve weeks. Each soft gelatin capsule contains a dry extract from *Silybum marianum* (151.5 mg, equivalent to 45 mg of silybins) [120]. We assessed adverse effects during each of the seven visits. Patients were instructed not to modify their eating or physical activity habits, not to ingest antioxidant multivitamin supplements or NSAIDs throughout the treatment and asked to report any changes in diet or physical activity.

### 4.3. Diet and Physical Activity Questionnaires

We obtained clinical nutritional history, 24 h dietary recall (24HR), habitual diet, food frequency, and validated physical activity questionnaires from enrolled patients at the beginning of our study from a clinical nutritionist to avoid significant variations in diet and physical activity [121,122,123]. Then, we saw patients every two weeks during six follow-up visits and obtained 24HR at every visit to monitor changes in diet.

### 4.4. Anthropometric Measurements and Determination of Insulin Resistance

We measured patient height with a stadiometer (Seca, model 206, Hamburg, Germany) on the first visit according to Lohman’s technique [124]. On the first and six subsequent visits, weight (kg), body fat (%), visceral fat (%), skeletal muscle (%), and BMI measurements were obtained from each patient with a bioimpedance scale (Omron Healthcare, model HBF-514C, Lake Forest, IL, USA). Waist and hip circumferences were measured according to Lohman’s technique [124]. Insulin sensitivity was estimated using the HOMA-IR (units of mass) [100].

### 4.5. Blood Sampling for Oral Glucose Tolerance Test

All patients underwent an OGTT before initiating and after completing 12 weeks of silymarin treatment. For the OGTT, fasting blood samples were obtained from all patients and then asked to ingest 75 g of an anhydrous glucose solution immediately after. Then, we collected blood samples from each patient at 30, 60, 90, and 120 min.

### 4.6. Biochemical Markers

We obtained blood chemistry analyses, lipid profiles, and determination of insulin levels from the blood samples of all patients, processed at the Hospital General de Mexico, Dr. Eduardo Liceaga laboratory service.

### 4.7. Statistical Analysis

We analyzed the normality of the data with the Kolmogorov–Smirnov test. Then, we analyzed datasets that passed normality tests by measuring changes in the same patient before (t = 0 weeks) and after silymarin treatment (t = 12 weeks). We evaluated the data with the Student’s *t*-test for paired samples, with a 95% CI. *p* values < 0.05 were considered statistically significant. We performed all tests using GraphPad Prism^®^ 5.0 software (GraphPad Software, Boston, MA, USA).

## 5. Conclusions

Silymarin treatment alone significantly reduced glucose and HOMA-IR levels in a group of IR non-diabetic obese women after twelve weeks of treatment, without standard treatment, dietary, or physical activity intervention. The results of our study support the benefits of silymarin in obese individuals with IR. However, more clinical studies in IR-obese individuals with silymarin treatment and a larger sample size are needed to strengthen the results of this study.

## Figures and Tables

**Table 1 ijms-25-02050-t001:** Anthropometric and body composition characteristics of enrolled patients at baseline and at 12 weeks of silymarin treatment.

Parameters	Baseline (*n* = 6)	At 12 Weeks (*n* = 6)	Difference	*p* *
Age (years)	34.4 ± 8.6	-		
Height (cm)	159.2 ± 8.4	-		
Weight (kg)	82.9 ± 11.6	82.9 ± 11.3	0.02	0.966
BMI (kg/m^2^)	33.9 ± 2.6	33.9 ± 2.2	0.00	1.000
Wc (cm)	91.8 ± 4.7	92.3 ± 9.9	0.42	0.917
Hc (cm)	112.7 ± 11.8	113.3 ± 8.8	0.60	0.843
TBF (%)	47.9 ± 1.7	47.2 ± 1.3	−0.77	0.163
Visc fat (%)	9.0 ± 1.4	8.8 ± 1.6	−0.17	0.363
Lean mass (%)	22.8 ± 1.4	23.4 ± 1.3	0.55	0.111

Data are expressed as the mean ± standard deviation and compared using Student’s *t*-test (*n* = 6). * *p* < 0.05 was considered statistically significant. Abbreviations: BMI, body mass index; Wc, waist circumference; Hc, hip circumference; TBF, total body fat; Visc fat: visceral fat.

**Table 2 ijms-25-02050-t002:** Glucose variation, lipid profile, liver function, inflammation markers of enrolled patients at baseline and at 12 weeks of silymarin treatment.

Parameters	Baseline (*n* = 6)	At 12 Weeks (*n* = 6)	Difference	*p*
FPG (mg/dL)	96.7 ± 4.2	91.2 ± 6.8	−5.5	0.041 *
FPI (mcU/mL)	14.9 ± 5.2	12.6 ± 3.8	−2.3	0.159
HOMA-IR	3.6 ± 1.2	2.8 ± 0.9	−0.7	0.048 *
Hb1Ac (%)	5.8 ± 0.3	5.8 ± 0.4	0	0.855
TC (mg/dL)	170.3 ± 33.9	161.3 ± 24.7	−9	0.393
LDL-C (mg/dL)	116.3 ± 28.7	110.2 ± 18.5	−6.1	0.554
HDL-C (mg/dL)	43.2 ± 8.8	41.8 ± 5.3	−1.4	0.500
Tg (mg/dL)	154.8 ± 47.6	145.0 ± 43.5	−9.8	0.198
AST (U/L)	19.7 ± 9.1	21.3 ± 9.5	1.6	0.437
ALT (U/L)	24.8 ± 21.4	28.3 ± 23.2	3.5	0.461
UA (mg/dL)	4.2 ± 0.6	4.2 ± 0.8	0	1.000
TB (mg/dL)	0.6 ± 0.1	0.6 ± 0.2	0	0.745
IB (mg/dL)	0.5 ± 0.1	0.5 ± 0.3	0	0.727
DB (mg/dL)	0.1 ± 0	0.2 ± 0.1	0.1	0.312
CRP (mg/dL)	3.4 ± 0.8	3.3 ± 1.2	−0.1	0.795

Data are expressed as the mean ± standard deviation and compared using Student’s *t*-test (*n* = 6). * *p* < 0.05 was considered statistically significant. Abbreviations: FPG, fasting plasma glucose; FPI, fasting plasma insulin; HOMA-IR, Homeostatic Model of the Insulin Resistance Index; Hb1Ac, glycated hemoglobin; TC, total cholesterol; LDL-C, low-density lipoprotein cholesterol; HDL-C, high-density lipoprotein cholesterol; Tg, triglycerides; AST, aspartate aminotransferase; ALT, alanine aminotransferase; UA, uric acid; TB, total bilirubin; IB, indirect bilirubin; DB, direct bilirubin; CRP, C-reactive protein.

**Table 3 ijms-25-02050-t003:** Differences in biochemical parameters of enrolled patients at baseline and after silymarin treatment (t = 12 weeks) during the oral glucose tolerance test.

		Fasting	Postprandial			
	Time (min)	0	30	60	90	120
Glucose (mg/dL)	Baseline	96.7 ± 4.2	147.2 ± 25.5	176.2 ± 36.1	155.4 ± 30.8	147.0 ± 21.5
	At 12 weeks	91.2 ± 6.8	144.6 ± 32.0	163.0 ± 15.1	146.8 ± 23.4	138.6 ± 26.3
	Difference	5.5	−2.6	−8.4	−4.6	−3.8
	*p*	0.041 *	0.867	0.498	0.715	0.687
Insulin (mcU/mL)	Baseline	14.9 ± 5.2	78.9 ± 34.7	139.5 ± 68.6	111.3 ± 62.6	150.1 ± 76.4
	At 12 weeks	12.6 ± 3.8	71.6 ± 37.3	109.8 ± 40.4	130.4 ± 44.1	120.9 ± 47.4
	Difference	−2.3	−7.3	−29.7	19.1	−29.2
	*p*	0.159	0.602	0.196	0.592	0.3

Data are expressed as the mean ± standard deviation and compared using Student’s *t*-test (*n* = 6). * *p* < 0.05 was considered statistically significant.

## Data Availability

We included the original contributions presented in our study in the article; please direct further inquiries to the corresponding authors.

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
