# Peer review of "Silymarin Reduced Insulin Resistance in Non-Diabetic Women with Obesity"

_ijms, 2024, doi:10.3390/ijms25042050_

Round 1
Reviewer 1 Report
Comments and Suggestions for Authors
Revision Report
The article entitled “Silymarin reduced insulin resistance in non-diabetic women 2 with obesity” is interesting and very informative. It will improve the scientific knowledge and treatment of insulin resistance. However, I have some queries that should be answered before its final decision.
What was the objective of study? Usually insulin resistance is studied in diabetic patients.
There are abbreviations used in the abstract without explanation.
The introduction is too much lengthy.
Last paragraph of introduction is very long.
Figure 1 is repletion of Table 2. It can be provided as supplementary
Same for Table 3 and Figure 2.
On what basis, exclusion criteria were set?
Comments on the Quality of English LanguageMinor corrections
Reviewer 2 Report
Comments and Suggestions for Authors
Dear Editor,
I carefully read the manuscript "Silymarin reduced insulin resistance in non-diabetic women with obesity".
My comments and suggestions for the authors are the following:
- Lines 117-121: The authors should more properly refer to a meta-analysis of RCTs testing the metabolic effect of berberine-silymarin association (doi: 10.1002/ptr.6282). Moreover, the authors should consider to refer to doi: 10.3389/fendo.2022.1089938 and doi: 10.5114/amsad/166571.
- All the abbreviations should be defined at their first occurrence in the manuscript.
- In the manuscript, the authors refer to LDL and HDL. However, I suppose they wanted to refer to cholesterol fractions instead of lipoproteins. Then, LDL and HDL should be replaced by LDL-C and HDL-C, respectively.
- Table 3: "Glucose" should be replaced by "Fasting plasma glucose".
- English language needs to be carefully revised and improved.
- The references should be formatted following the Instructions for the Authors.
- Statistical analysis should be more properly described.
- The limitations of the study should be further and more deeply discussed by the authors.
- The manuscript is not balanced in its parts. The authors included in the Introduction information that should be more properly included in the Discussion.
- The main critical issue is the lack of a control group. The presence of a control group is critical to claim that silymarin may offer a therapeutic alternative to improve IR in non-diabetic individuals with obesity. I think that authors' conclusions are too speculative.
Comments on the Quality of English LanguageThe manuscript is poorly written. An extensive revision is needed.
Round 2
Reviewer 2 Report
Comments and Suggestions for Authors
Dear Editor,
I carefully read the revised version of the manuscript, that is improved when compared with the original version.